# CONSEQUENTIALIST CONDITIONAL COOPERATION IN SOCIAL DILEMMAS WITH IMPERFECT INFORMATION

**Alexander Peysakhovich & Adam Lerer** [*]
Facebook AI Research
New York, NY
{alexpeys,alerer}@fb.com

## ABSTRACT

Social dilemmas, where mutual cooperation can lead to high payoffs but participants face incentives to cheat, are ubiquitous in multi-agent interaction. We wish to construct agents that cooperate with pure cooperators, avoid exploitation by pure defectors, and incentivize cooperation from the rest. However, often the actions taken by a partner are (partially) unobserved or the consequences of individual actions are hard to predict. We show that in a large class of games good strategies can be constructed by conditioning one's behavior solely on outcomes (ie. one's past rewards). We call this consequentialist conditional cooperation. We show how to construct such strategies using deep reinforcement learning techniques and demonstrate, both analytically and experimentally, that they are effective in social dilemmas beyond simple matrix games. We also show the limitations of relying purely on consequences and discuss the need for understanding both the consequences of and the intentions behind an action.

## 1 INTRODUCTION

Deep reinforcement learning (RL) is concerned with constructing agents that start as blank slates and can learn to behave in optimal ways in complex environments.[1] A recent stream of research has taken a particular interest in social dilemmas, situations where individuals have incentives to act in ways that undermine socially optimal outcomes (Leibo et al., 2017; Perolat et al., 2017; Lerer & Peysakhovich, 2017; Kleiman-Weiner et al., 2016). In this paper we consider RL-based strategies for social dilemmas in which information about a partner's actions or the underlying environment is only partially observed.

The simplest social dilemma is the Prisoner's Dilemma (PD) in which two players choose between one of two actions: cooperate or defect. Mutual cooperation yields the highest payoffs, but no matter what one's partner is doing, one can get a higher reward by defecting. A well studied strategy for maintaining cooperation when the PD is repeated is tit-for-tat (TFT, Axelrod (2006)). TFT behaves by copying the prior behavior of their partner, rewarding cooperation today with cooperation tomorrow. Thus, if an agent commits to TFT it makes cooperation the best strategy for the agent's partner. TFT has proven to be a heavily studied strategy because it has intuitive appeal: 1) it is easily explainable, 2) it begins cooperating, 3) it rewards a cooperative partner, 4) it avoids being exploited, 5) it is forgiving.

In Markov games cooperation and defection are not single actions, but rather temporally extended policies. Recent work has considered expanding TFT to more complex Markov games either as a heuristic, by learning cooperative and selfish policies and switching between them as needed (Lerer & Peysakhovich, 2017), or as an outcome of an end-to-end procedure (Foerster et al., 2017c). TFT is

---

[*]Both authors contributed equally to this paper. Author ordering was determined at random.

[1]This approach has been applied to domains including: single agent decision problems (Mnih et al., 2015), board and card-based zero-sum games (Tesauro, 1995; Silver et al., 2016; Heinrich & Silver, 2016), video games (Kempka et al., 2016; Wu & Tian, 2016; Ontanón et al., 2013; Usunier et al., 2016; Foerster et al., 2017a), multi-agent coordination problems (Lowe et al., 2017; Foerster et al., 2017b; Riedmiller et al., 2009; Tampuu et al., 2017; Peysakhovich & Lerer, 2017), and the emergence of language (Lazaridou et al., 2017; Das et al., 2017; Evtimova et al., 2017; Havrylov & Titov, 2017; Jorge et al., 2016).

an example of a *conditionally cooperative* strategy - that is, it cooperates when a certain condition is fulfilled (ie. the partner's last period action was cooperative). TFT, however, has a weakness - it requires perfect observability of a partner's behavior and perfect understanding of each action's future consequences.

Our main contribution is to use RL methods to construct conditionally cooperative strategies for games with imperfect information. When information is imperfect, the agent must use what they can observe to try to estimate whether a partner is acting cooperatively (or not) and determine how to respond. We show that when the game is ergodic, observed rewards can be used as a summary statistic - if the current total (or time averaged) reward is above a time-dependent threshold (where the threshold values are computed using RL and a form of self play) the agent cooperates, otherwise the agent does not[2]. We call this consequentialist conditional cooperation (CCC). We show analytically that this strategy cooperates with cooperators, avoids exploitation, and guarantees a good payoff to the CCC agent in the long run.

We study CCC agents in a partially observed Markov game which we call Fishery. In Fishery two agents live on different sides of a lake in which fish appear. The game has partial information because agents cannot observe what happens across the lake. Fish spawn randomly, starting young and swim to the other side and become mature. Agents can catch fish on their side of the lake. Catching any fish yields payoff but mature fish are worth more. Therefore, cooperative strategies are those which leave young fish for one's partner. However, there is always a temptation to defect and catch both young and mature fish. We show that CCC agents cooperate with cooperators, avoid exploitation, and get high payoffs when matched with themselves.

Second, we show that CCC is an efficient strategy for more complex games where implementing conditional cooperation by fully modeling the effect of an action on future rewards (eg. amTFT (Lerer & Peysakhovich, 2017)) is computationally demanding. We compare the performance of CCC to amTFT in the Pong Player's Dilemma (PPD). This game is a modification of standard Atari pong such that when an agent scores they gain a reward of $1$ but the partner receives a reward of $-2$. Cooperative payoffs are achieved when both agents try hard not to score but selfish agents are again tempted to defect and try to score points even though this decreases total social reward. We see that CCC is a successful, robust, and simple strategy in this game.

However, this does not mean CCC completely dominates forward looking strategies like amTFT. We consider a version of the Pong Players' Dilemma where when a player scores, instead of their partner losing 2 points deterministically they lose $2/p$ points with probability $p$. Here the *expected* rewards of non-cooperation are the same as in the PPD and so expected-future-reward based methods (eg. amTFT) will act identically. However, when $p$ is low it may take a long time for consequentialist agents to detect a defector. Empirically we see that in short risky PPD games CCC agents can be exploited by defectors but that amTFT agents cannot. We close by discussing limitations and progress towards agents that can effectively use both intention and outcome information effectively in navigating the world.

## 1.1 RELATED WORK

Game theorists have studied the emergence of cooperation in bilateral relationships under both perfect and imperfect observation (Green & Porter, 1984; Fudenberg & Maskin, 1986; Fudenberg et al., 1994; Axelrod, 2006; Kamada & Kominers, 2010; Abreu et al., 1990). However, this research program almost exclusively studies repeated matrix games and focuses mostly on proving the existence of equilibria which maintain cooperation rather than on constructing simple strategies that do well across many complex situations. Other work has constructed algorithms for explicitly computing these folk theorem strategies (Littman & Stone, 2005; de Cote & Littman, 2008) but it focuses on perfectly observed games played iteratively rather than imperfectly observed games played once at test time.

In addition, the question of designing a good agent for social dilemmas can sometimes be quite different from questions about computing equilibrium strategies. For example, in the repeated PD, tit-for-tat is held up as a good strategy for an agent to commit to (Axelrod, 2006). However, both players using tit-for-tat is not an equilibrium (since the best response to tit-for-tat is always cooperate).

---

[2]In an ideal world we may want to construct a full posterior using Bayesian methods. However, this is often computationally difficult in practice.

A related literature on multi-agent learning focuses on studying how agent properties (learning rules, game parameters, etc...) affect the dynamics of behavior (Fudenberg & Levine, 1998; Sandholm & Crites, 1996; Shoham et al., 2007; Nowak, 2006; Conitzer & Sandholm, 2007; Leibo et al., 2017; Perolat et al., 2017). A related set of work looks at learning can be shaped to converge to better outcomes (Babes et al., 2008). These works study questions related to ours, in particular, designing agents which 'teach' their learning partners (Foerster et al., 2017c). However they deals with a different setup (more than a single game played at test time). In addition, these techniques may require detailed knowledge of the game structure (to eg. construct reward shaping as in Babes et al. (2008)) or a partner's updating rule (as in Foerster et al. (2017c)). An interesting direction for future work is to blend the learning approaches with the trigger strategy approach we study here.

## 2 CONSEQUENTIALIST CONDITIONALLY COOPERATIVE STRATEGIES

We work with partially observed Markov games (POMG), which are multi-agent generalizations of partially observed Markov decision problems:

**Definition 1** *A (two-player, finite)* **partially observed Markov game** *(POMG) consists of: a finite set of states $S$; a set of actions for each player $\mathcal{A}_1, \mathcal{A}_2$; a transition function $\tau : S \times \mathcal{A}_1 \times \mathcal{A}_2 \to \Delta(S)$; an observation function that tells us what each player observes $_i : S \times \mathcal{A}_1 \times \mathcal{A}_2 \to \Delta(\Omega_i)$ where $\Omega_i$ is a set of possible observations; and a reward function that maps states and actions to each player's reward $R_i : S \times \mathcal{A}_1 \times \mathcal{A}_2 \to \Delta(\mathbb{R})$.*

We assume that per turn rewards are bounded above and below. Agents choose a policy

$$\pi_i : \Omega_i \to \Delta(\mathcal{A}_i)$$

which takes as input the observation and outputs a probability distribution on actions - this is similar to the notion of 'belief free strategies' in the study of repeated games (Ely et al., 2005). Given a pair of policies, one for each agent, and a starting state, we define each player's value function as the average (undiscounted) reward if players start in state $s$ and follow policies $\pi_1, \pi_2$, i.e.

$$V_i(s, \pi_1, \pi_2) = \lim_{t \to \infty} E\left[\frac{1}{t} \sum_{k=0}^{t} r_k^i\right]$$

Note that while we will prove results in the undiscounted setting, for $\delta$ sufficiently close to 1, the optimal policy is the same in the undiscounted and discounted setting, so standard discounted policy gradient techniques can still be used (Schwartz, 1993).

We restrict ourselves to reward-ergodic POMGs:

**Assumption 1 (Reward Ergodicity in POMG)** *Say a POMG is reward-ergodic if for any pair of policies $\pi_1, \pi_2$ the long-run reward has a well defined rate independent of the starting state almost surely. Formally, this means for any starting state $s$, and either player $i$, there exists a rate $\rho_{\pi_1 \pi_2}^i$ such that*

$$\lim_{t \to \infty} V_i(s, \pi_1, \pi_2) \xrightarrow{a.s.} \rho_{\pi_1 \pi_2}^i$$

Any pair of policies applied to a POMG creates a Markov chain of underlying states. If for any pair of policies that Markov chain is 'unichain', that it, it has a single positive recurrent chain (Puterman, 2014) then the POMG will be reward ergodic (Meyn & Tweedie, 2012)[Thm. 17.0.1]. The unichain assumption is often used in applications of RL methods in the undiscounted RL problem (Schwartz, 1993).

**Definition 2** *Cooperative Policies are those that maximize the joint rate of reward:*

$$(\pi_1^C, \pi_2^C) \in argmax_{\Pi_1, \Pi_2}(V_1(\pi_1, \pi_2) + V_2(\pi_1, \pi_2)).$$

*Let $\Pi^C$ be the set of such tuples.*

We look at the class of POMGs which have two restrictions:

**Assumption 2 (Social Dilemma)** *For any player $i$ and any $(\pi_1^C, \pi_2^C) \in \Pi^C$ we have that*

$$\pi_i^C \notin argmax_{\Pi_i}(V_i(s, \pi_i, \pi_j^C)).$$

**Assumption 3 (Exchangeability of Cooperative Strategies)** *For any $(\pi_1, \pi_2) \in \Pi^C$ and $(\pi_1', \pi_2') \in \Pi^C$ we have that $(\pi_i', \pi_j) \in \Pi^C$.*

Note that if this exchangeability assumption is not satisfied we have both a cooperation problem (should agents defect?) and also a coordination problem (if we choose to cooperate, how do we choose among the multiple potential ways to cooperate?) We point the reader to Kleiman-Weiner et al. (2016) for further discussion of this issue. Solving the coordination problem (eg. by introducing communication) is beyond the scope of this paper though it is an important avenue for future work.

To construct a CCC agent we need access to a policy pair $(\pi_1^D, \pi_2^D)$ that forms a Nash equilibrium of the game. We assume that these strategies generalize two properties of defection in the Prisoner's Dilemma: 1) the have lower rates of payoff than the socially optimal strategies in the long run for both players and 2) if we take a mixed policy $\pi^{C+D}$ which behaves according to $\pi^C$ at some periods and $\pi^D$ at others then $V_i(\pi_i^{C+D}, \pi_j^C) \geq V_i(\pi_i^C, \pi_j^C)$. This last condition is essentially saying that $\pi^D$ is a selfish policy that, even if used some of the time, still increases the payoffs of the player choosing it (while decreasing total social efficiency). We explain a weaker condition in the appendix.

To behave according to CCC our agent maintains a persistent state at each time period $C_t^i$ which is the current time-averaged reward it has received.

Given a threshold $T$, the agent plays according to $\pi^C$ if $C_t^i > T$ and $\pi^D$ otherwise. Let $\rho_{CC}$ be the rate associated with both players behaving according to $\pi^C$ and let $\rho_{CD}$ be the rate associated with our agent playing according to $\pi^C$ and the other agent behaving according to $\pi^D$. Let $T = (1-\alpha)\rho_{CC} + \alpha\rho_{CD}$, where $0 < \alpha < 1$ is a slack parameter that specifies the agent's leniency. We present the following result:

**Theorem 1** *Consider a strategy where agent $1$ acts according to $\pi_1^C$ if $C_t^i > T$ and $\pi_1^D$ otherwise. This gives two guarantees in the long run:*

1. ***Cooperation Wins:** If agent $2$ acts according to $\pi_2^C$ then for both agents $\lim_{t\to\infty} \frac{1}{t}\sum r_t^i = \rho_{CC}^i$.*

2. ***Defecting Doesn't Pay:** If agent $2$ acts according to a policy that gives agent $1$ a payoff of less than $T$ in the long run then $\lim_{t\to\infty} \frac{1}{t}\sum r_t^2 \leq \rho_{DD}^2 < \rho_{CC}^2$.*

Thus, a simple threshold based strategy for player 1 makes cooperation a better strategy than defection in the long-run for player 2 in any ergodic game. CCC satisfies the desiderata we set out at the beginning: it is simple to understand, cooperates with cooperators, does not get exploited by pure defectors, incentivizes rational partners to cooperate, and, importantly, gives a way to return to cooperation if it has failed in the past.[3]

The CCC strategy also provides a payoff guarantee against rational learning agents: if one's partner is a learning agent who best responds in the long-run then, since $\pi^D$ forms an equilibrium, a CCC agent can always guarantee themselves a payoff of at least $\rho_{DD}$ in the long-run. Unfortunately, CCC does not give any adversarial guarantees in general. Extending conditionally cooperative strategies to be robust to not only selfish agents trying to cheat but adversarial agents trying to actively destroy value is an interesting direction for future work.

However, we note that the simple construction above only gives long-run guarantees. We now focus on generalizing this strategy to work well in finite time as well as how to use RL methods to compute its components. Finally, we note that this strategy can be extended to some non-payoff ergodic games. For example, if there is a persistent but unknown state which affects both rewards equally (say, multiplies them by some factor) then the amount of inequality (eg. the ratio or difference of rewards or more complex function such as those used by Fehr & Schmidt (1999)) can be used as a summary statistic.

---

[3]Cooperation returns when CCC returns to a time averaged payoff above the threshold, this means a partner can accelerate this process by taking actions to give the CCC agent extra payoff. We refer to this process as 'giving flowers to apologize.'

## 3    RL IMPLEMENTATION OF CCC

To construct the $\hat{\pi}^C$ and $\hat{\pi}^D$ policies used by CCC, we follow the training procedure of Lerer & Peysakhovich (2017). We perform self-play with modified reward schedules to compute the two policies:

1. Selfish - here both players get rewards equal to their own reward. This is the standard self-play paradigm used in multi-agent zero-sum settings. We refer to the learned policies here as $\hat{\pi}^D$

2. Prosocial - here both players get rewards at each time step not only for their own reward, but also for the reward their partner receives. We refer to the learned polices as $\hat{\pi}^C$

Our agents are set up as standard deep RL agents which take game state (e.g. pixels) as input and pass them through a convolutional neural network to compute a distribution over actions. The architectures of the agents as well as the training procedures are standard and we put them in the appendix.

We note that learning policies via RL in POMDPs has unique challenges (Jaakkola et al., 1995). The correct choice of learning algorithm will depend on the situation; policy gradient is preferred for POMDPs because optimal policies may be nondeterministic, and a common approach is to perform a variant of policy gradient with a function approximator augmented with an RNN 'memory' that keeps track of past states (Heess et al., 2015). In our Fishery game, the policy (but not the value) is independent of the unobserved state, so RNN augmentation was unnecessary; however, since our policy was stateless we had to avoid value-based methods (e.g. actor-critic) because the aliasing of *values* prevents these methods from finding good policies.

Having computed the policies, we need to compute thresholds for conditional cooperation. There are 3 important sources of finite time variance that a threshold needs to account for: first, a partner's $\pi^C$ may not be the same as our agent's due to function approximation. Second, initial states of the game may matter in finite time. Third, there may be inherent stochasticity in the rewards.

We compute the per-turn threshold $\hat{T}^i_t$ as follows: first we take our learned $\hat{\pi}^C$ and $\hat{\pi}^D$ and perform $k$ rollouts of the full game assuming cooperate (we call the resulting per period cumulative payoffs to our agent $\hat{R}^{tk}_{CC}$ where $k$ corresponds to the iteration and $t$ to the time). We also compute batches of rollouts of a paired cooperator and defector $\hat{R}^{tk}_{CD}$. We let $\bar{R}^t_{CC}$ be the bottom $q^{th}$ percentile of these sample paths and we define our time dependent threshold in terms of cumulative reward as

$$\hat{T}^t = (1-\alpha)\bar{R}^t_{CC} + \alpha\frac{1}{k}\sum_k R^{tk}_{CD}.$$

If the CCC agent's current cumulative reward is above $\hat{T}^t$ they behave according to $\hat{\pi}^C$, otherwise they use $\hat{\pi}^D$.

This process gives us slack to account for the three sources of error described above. Tuning the parameters $q, \alpha$ allows us to trade off between the importance of false positives (detecting defection when one hasn't occurred) and false negatives (missing a defecting opponent). The algorithm is formalized into pseudocode below. We also show an example of threshold computation as well as associated precision/recall with actual opponents in the example below.

## 4    EXPERIMENTS

### 4.1    EXPERIMENT: FISHERY

Our first example is a common pool resource game which we call Fishery. In Fishery two agents live on different sides of a lake in which fish appear. Each side of the lake is instantiated as a $5 \times 5$ grid and agents can walk in all cardinal directions. Fish spawn randomly, starting young and swim to the other side and become mature. Agents can catch fish on their side of the lake by walking over them. Catching young fish yields 1 reward while mature fish yield a reward of 3.

In Fishery cooperative strategies are those which leave young fish for one's partner. However, there is always a temptation to defect and catch both young and mature fish. Fishery is an imperfect

---

**Algorithm 1** CCC as Agent 1

---

$\quad$**Input:** $\hat{\pi}^C, \hat{\pi}^D, \alpha, q, k$
$\quad$**for** $b$ in $range(0, k)$ **do**
$\quad\quad$ $s_{CC}[b] \leftarrow NewGame()$
$\quad\quad$ $s_{CD}[b] \leftarrow NewGame()$
$\quad$**while** $Game$ **do**
$\quad\quad$**for** $b$ in $range(0, k)$ **do**
$\quad\quad\quad$ $s_{CC}[b], R_{CC}[b] \leftarrow Step(s_{CC}[b], \hat{\pi}_1^C, \hat{\pi}_2^C)$ $\qquad$ ▷ Step returns next state and total reward
$\quad\quad\quad$ $s_{CD}[b], R_{CD}[b] \leftarrow Step(s_{CD}[b], \hat{\pi}_1^C, \hat{\pi}_2^D)$
$\quad\quad$ $\bar{R}_{CC} \leftarrow quantile(R_{CC}, q)$
$\quad\quad$ $\bar{R}_{CD} \leftarrow mean(R_{CD})$
$\quad\quad$ $T \leftarrow (1 - \alpha)\bar{R}_{CC} + \alpha \bar{R}_{CD}$
$\quad\quad$**if** $CurrentTotalReward < T$ **then**
$\quad\quad\quad$ Choose $a = \hat{\pi}_1^D(o)$
$\quad\quad$**else**
$\quad\quad\quad$ Choose $a = \hat{\pi}_1^C(o)$

---

observation game because agents cannot see the behavior of their partners across the lake. Figure 1 shows an example of a threshold in our Fishery game with $q = .1$ and $\alpha = .05$, we see that CC trajectories remain mostly above the threshold (meaning low false positives) and CD trajectories mostly lie below even after a short time period (meaning low false negatives). The game as well as the experimental results are shown in Figure 1.

We train 50 pairs of agents under the selfish and prosocial reward schemes using a policy gradient method with simple CNN policy approximators (see Appendix for details). We see that selfish training leads to agents that defect and choose greedy, suboptimal strategies whereas prosocial training finds good policies. We then compute thresholds as described above and implement CCC agents.

First, we construct 22 matchups between CCC and pure cooperator and pure defecting partners. We see that CCC agents quickly avoid full exploitation by defectors while maintaining cooperation with cooperators (Figure 1 panel D). To see whether CCC satisfies the desiderata we laid out in the introduction we consider a tournament where we draw random policies from our trained pool of 50 and have them play a 1000 time step Fishery game (we use fixed lengths to normalize the payoffs across games).

To compare strategies in the tournament and see how well they achieve our desiderata, we adopt the metrics from Lerer & Peysakhovich (2017). Let $S_i(X, Y)$ be the expected reward to player $i$ when a policy of type $\pi_1^X$ are matched with type $\pi_2^Y$. SelfMatch$(X) = S_1(X, X)$ measures whether a strategy achieves good outcomes with itself; Safety$(X) = S_1(X, D) - S_1(D, D)$ measures how a strategy is safe from exploitation by a defector; and IncentC$(X) = S_2(X, C) - S_2(X, D)$ measures whether a strategy incentivizes cooperation from its partner. A full matrix of how well each policy does against each other policy is provided in the Appendix.

## 4.2 EXPERIMENT: PONG PLAYERS' DILEMMA

Since any perfectly observed game is trivially a partially observed game CCC can also be used in games of perfect information. We consider the Pong Players' Dilemma (PPD) which has been used in prior work to evaluate perfect information forward-looking conditionally cooperative strategies Lerer & Peysakhovich (2017).The PPD alters the reward structure of Atari Pong so that whenever an agent scores a point they receive a reward of 1 and the other player receives $-2$ (Tampuu et al., 2017). In the PPD the only (jointly) winning move is not to play. However, selfish agents are again tempted to defect and try to score points even though this decreases total social reward.

We compare CCC to the forward-looking approximate Markov Tit-for-Tat (amTFT Lerer & Peysakhovich (2017)). amTFT conditions its cooperation on a counterfactual future reward - the amTFT agent sees the actions taken by their partner and, if the action is not the one suggested by $\pi^C$ uses the game's $Q$ function (which is learned or simulated via rollouts) to estimate the one shot-gain to the partner from taking this action. The agent keeps track of the total 'debit' a partner has accrued

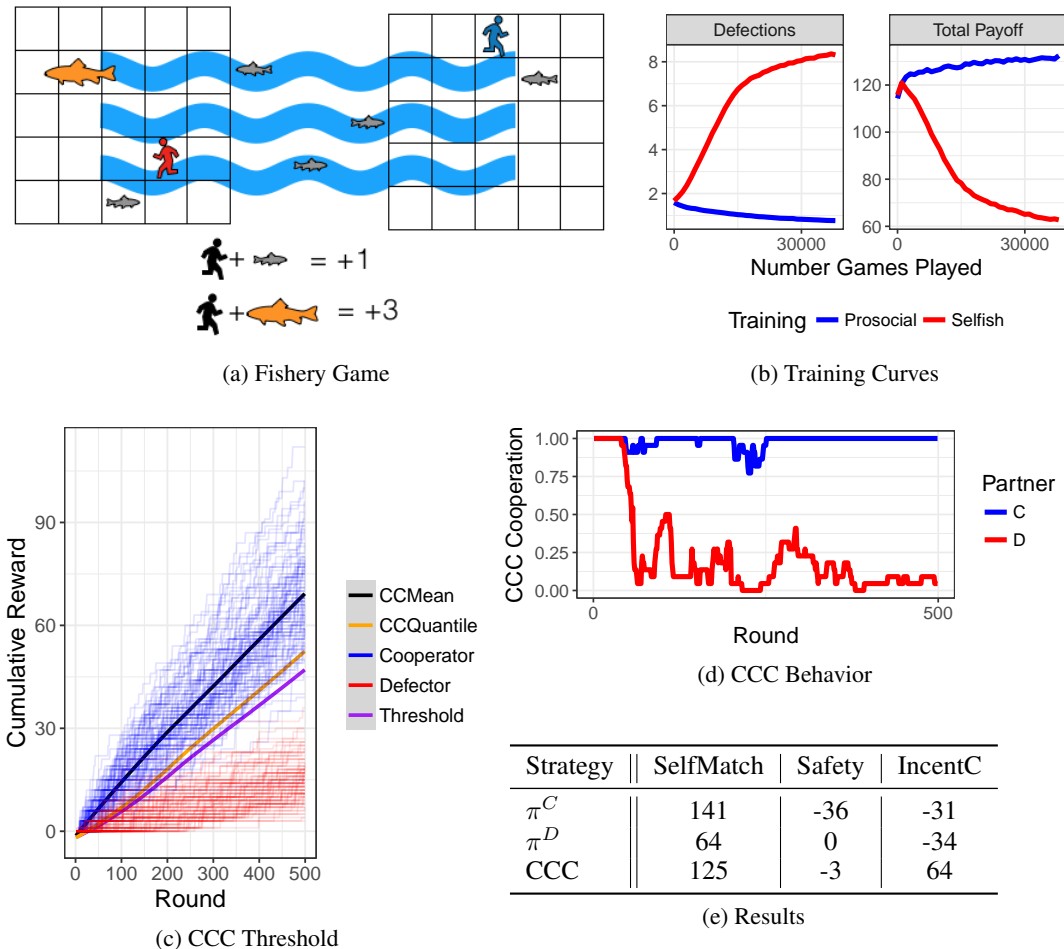

Figure 1: In Fishery two agents live on opposite sides of a lake and cannot observe each other's actions directly. Each time step fish can spawn on their side of the lake and begin to swim to the other side. Fish start young and become mature if they are allowed to enter the middle of the lake. Training using selfish self-play leads to agents that try to eat all the fish and thus cannot reach optimal payoffs, while social training finds cooperative strategies. Panel C shows example trajectories of payoffs as well as the $CCC$ per-round threshold. Panel D shows trajectories of behavior by CCC agents when faced with C or D partners. Panel E shows that CCC does well with itself, is not easily exploited, and incentivizes cooperation from its partners.

over time and if that crosses a threshold the amTFT agent then behaves according to $\pi^D$ for enough periods such that the partner's debit is wiped out. We call this type of strategy intention-based because it computes, at the time of the action, the expected future consequences rather than waiting for those consequences to occur as CCC agents do.

To make the comparison fair we use the 18 pairs of agents trained in Lerer & Peysakhovich (2017) by the selfish and prosocial reward schemes using randomly determined game lengths using a standard A3C implementation (Mnih et al. (2016); see Lerer & Peysakhovich (2017) for complete details). Selfish training leads to selfish agents that try hard to score every point while prosocial training leads to cooperative agents that hit the ball directly back and forth (Figure 2). We construct CCC agents as in the Fishery experiment above and see how CCC performs against $\pi^C$, $\pi^D$, and itself in fixed length PPD games. As with Fishery, we find that CCC cooperates with cooperators (and itself) but does not get exploited by defectors.

amTFT is more computationally expensive than CCC because it requires the use of a $Q$ function (which can be hard to train) or rollouts (which are expensive to compute) we follow the procedure in

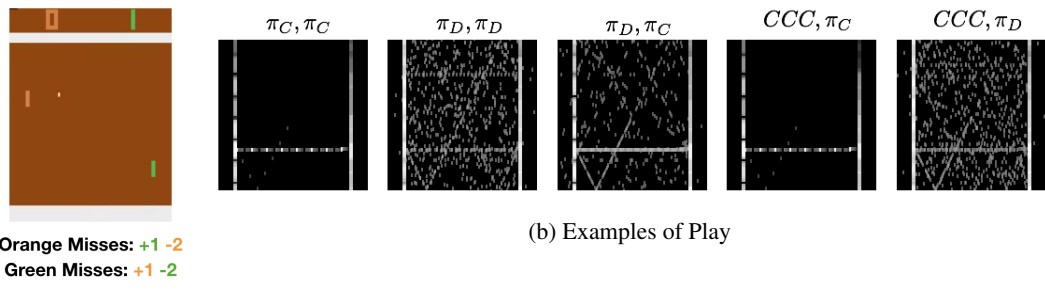

(b) Examples of Play

Orange Misses: +1 -2
Green Misses: +1 -2

(a) PPD

| Strategy | SelfMatch | Safety | IncentC |
|---|---|---|---|
| $\pi^C$ | 0 | -18.4 | -12.3 |
| $\pi^D$ | -5.9 | 0 | -18.4 |
| CCC | 0 | -4.6 | 3.3 |
| amTFT | -1.6 | -5.2 | 2.6 |

| Strategy | SelfMatch | Safety | IncentC |
|---|---|---|---|
| $\pi^C$ | -0.7 | -23.6 | -12.8 |
| $\pi^D$ | -5.8 | 0 | -22.6 |
| CCC | -0.2 | -12.2 | -5.7 |
| amTFT | -3.6 | -3.1 | 2.5 |

(c) Results (PPD)  (d) Results (Risky PPD)

Figure 2: In the Pong Player's Dilemma selfish training leads to agents that try hard to score and thus end up with bad payoffs. Cooperators learn to gently hit the ball back and forth. CCC agents behave like a cooperators when faced with cooperators and prevent themselves from being exploited by defectors. Panel B shows example PPD games between different strategies with brightness of a pixel indicating proportion of time ball spends in that location. Panels C and D present comparisons of strategies in the PPD and Risky PPD. In PPD, CCC achieves similar performance to the more expensive amTFT. In the finite length risky PPD, however, CCC loses both its ability to incentivize cooperation and to avoid exploitation.

Lerer & Peysakhovich (2017) to construct amTFT agents for the PPD. However in a tournament we see that CCC agents, which are much simpler, perform just as well in the PPD.

Does this mean that CCC completely dominates amTFT in perfectly observed games? The answer is no. In particular, intention-based strategies can be effective on much shorter timescales than CCC. To demonstrate this we further modify the reward structure of the PPD so that when a player scores, instead of their partner losing 2 points deterministically they lose $2/p$ points with probability $p$. Here the *expected* rewards of non-cooperation are the same as in the PPD and so amTFT acts similarly (though now we require large batches if using rollouts). However, we see that in intermediate length games (1000 time steps) and $p = .1$ CCC agents can be exploited by defectors.[4] A full table of how each strategy performs against each other strategy is available in the Appendix.

We also study CCC in another perfectly observed grid based social dilemma, Coins (Lerer & Peysakhovich, 2017; Foerster et al., 2017c). The results mirror those of the PPD so we relegate them to the Appendix.

## 5 CONCLUSION, LIMITATIONS AND FUTURE WORK

In this work we have introduced consequentialist conditionally cooperative strategies and shown that they are useful heuristics in social dilemmas, even in those where information is imperfect either due to the structure of the game or due to the fact that we cannot perfectly forecast a partner's future actions. We have shown that using one's own reward stream as a summary statistic for whether to cooperate (or not) in a given period is guaranteed to work in the limit as long as the underlying game is ergodic. Note that this sometimes (but not always) gives good finite time guarantees. In particular,

---

[4]For simplicity for this experiment we use the $\hat{\pi}^C$ and $\hat{\pi}^D$ strategies trained in the standard PPD for the risky PPD, this is because the risky PPD is the same (in expectation) as the PPD.

the time scale for a CCC agent to detect exploitation is related to the mixing time of the POMG and the stochasticity of rewards; if these are large, then correspondingly long games are required for CCC to perform well.

We have also compared consequentialist and forward-looking models. As another simple example of the difference between the two we can consider the random Dictator Game (rDG) introduced by Cushman et al. (2009). In the rDG, individuals are paired, one (the Dictator) is given an amount of money to split with a Partner, and chooses between one of two dice, a 'fair' die which yields a $50 - 50$ split with a high probability and an unfair split with a low probability and an 'unfair' die which yields a $50 - 50$ split with low probability. Consequentialist conditional cooperators would label a partner a defector if an unfair outcome came up (regardless of die choice) whereas intention-based cooperators would look at the choice of die, not the actual outcome.

For RL trained agents, conditioning purely on intentions (eg. amTFT) has advantages in that it is forward looking and doesn't require ergodicity assumptions but it is an expensive strategy that is complex (or impossible) to implement for POMDPs and requires very precise estimates of potential outcomes. CCC is simple, works in POMDPs and requires only information about payoff rates (rather than actual policies), however it may take a long time to converge. Each has unique advantages and disadvantages. Therefore constructing agents that can solve social dilemmas will require combining consequentialist and intention-based signals.

Interestingly, experimental evidence shows that while humans combine both intentions and outcomes, we often rely much more heavily on consequences than 'optimal' behavior would demand. For example, experimental subjects rely heavily on the outcome of the die throw rather than die choice in the rDG (Cushman et al., 2009). This is evidence for the notion that rather than acting optimally in each situation, humans have social heuristics which are tuned to work across many environments (Rand et al., 2014; Hauser et al., 2014; Ouss & Peysakhovich, 2015; Arechar et al., 2016; Mao et al., 2017; Niella et al., 2016). There is much discussion of hybrid environments that include both artificial agents and humans (eg. Shirado & Christakis (2017); Crandall et al. (2017)). Constructing artificial agents that can do well in such environments will require going beyond the kinds of optimality theorems and experiments highlighted in this and related work.

In addition, we have defined cooperative policies as those which maximize the sum of the rewards. This seems like a natural focal point in symmetric games like the ones we have studied but it is well known that human social preferences take into account factors such as inequity (Fehr & Schmidt, 1999) and social norms (Roth et al., 1991). To be successful, AI researchers will have to understand human social heuristics and construct agents that are in tune with human moral and social intuitions (Bonnefon et al., 2016; Greene, 2014).

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

## 6 TECHNICAL APPENDIX

### 6.1 PROOF OF MAIN THEOREM

We will use this basic property of almost sure convergence. If a sequence of random variables $X_n$ converges to $X$ almost surely then

$$\forall \delta > 0, \epsilon > 0 \; \exists n_0 \text{ s.t. } P(\forall n > n_0, |X_n - X| < \delta) > 1 - \epsilon$$

**Cooperation Wins:**

The intuition behind the proof is as follows: first, we show that if the CCC agent's partner plays $\pi^C$ then for any $R_s > 0$ there exists a time $t_s$ at which the CCC agent's total payoff exceeds the threshold $t_s T$ by at least $R_s$ with high probability. Intuitively this is because the rate $T$ is lower than $\rho_{CC}$ which is weakly lower than the rate guaranteed to a CCC agent whose partner behaves according to $\pi^C$ always.

Second, we will show that for sufficiently large $R_s$, if the CCC agent's total payoff exceeds the threshold by $R_s$ then the $CCC$ agent also only plays $\pi^C$ from that point on with high probability.

Together this implies that if the partner plays according to $\pi^C$ then with high probability the CCC agent behaves according to $\pi^D$ only a finite amount of times and thus the rates of payoffs for both agents converge to $\rho_{CC}$.

**(Part 1)**

Let $R_t$ be the total reward of the CCC agent at time $t$ and pick some $R_s$. We assumed that $\rho_{C+D,C} \geq \rho_{CC} > T$. Therefore if the partner plays $\pi_C$ then $\frac{R_t}{t} \to \rho_{CCC,C} > T$. We set $\delta = \frac{\rho_{CC}-T}{2}$ and use almost sure convergence to say that

$$\forall \epsilon > 0 \; \exists t_m \text{ s.t. } P\left(\forall t > t_m, \frac{R_t}{t} > T + \frac{\rho_{CC}-T}{2}\right) > 1 - \epsilon.$$

Let $t_s = \max\left(t_m, \frac{R_s}{(\rho_{CC}-T)/2}\right)$. Then after $t_s$ turns the CCC agents total reward will exceed $t_s T$ by at least $R_s$ with probability at least $1 - \epsilon$.

**(Part 2)**

If we assume that both players play $\pi^C$ forever then by reward ergodicity and the fact that $\rho_{CC} > T$ we have

$$\forall \epsilon > 0 \; \exists t_0 \text{ s.t. } P\left(\forall t > t_0, \frac{R_t}{t} > T\right) > 1 - \epsilon.$$

Assuming bounded rewards $|r| \leq r_{max}$ then CCC agent's accumulated reward in $t_0$ turns is no less than $-t_0 r_{max}$. So, if at time $t$ the CCC agent has total payoff of $t_0 r_{max} + tT$, then with probability $1 - \epsilon$, their payoff will never fall below the threshold after that point and thus they will **always** play $\pi^C$. This validates the initial assumption for Part 2 that both players play $\pi^C$.

Putting it all together, setting $R_s$ to $t_0 R_{max}$ we have that for any $\epsilon > 0$ there exists a $t_s$ such that $R_{t_s} - t_s T \geq t_0 R_{max}$ with probability at least $1 - \epsilon$, and given that initial balance then both players play $\pi^C$ from that point forward and get average rewards according to $\rho_{CC}$, with probability $1 - \epsilon$. Therefore, with probability at least $1 - 2\epsilon$, both players will achieve average reward of $\rho_{CC}$.

**Defecting Doesn't Pay:**

Suppose agent 1's average reward converges to $\rho < T$. Then for some $t_0$ we have that $\frac{R_t}{t} < T$ for all $t > t_0$, with probability 1. Starting at $t_0$, CCC plays a fixed (stateless) policy $\pi_1^D$. Since $\pi^D$ forms an equilibrium agent 2 can achieve average reward of at most $\rho_{DD}$.

### 6.2 A NOTE ON $\pi^{C+D}$ PAYOFFS

While constructing CCC strategies, we make the assumption that the payoff of the pair $(\pi^{C+D}, \pi^C)$ to agent 1 is at least $\rho_{CC}$. This is an overly restrictive condition, because it precludes methods of

cooperation that take more than one period to execute. For example, if cooperation is walking to the left side of the room and defection is walking to the right side of the room, then a mix of the two policies may leave me in the middle gaining no reward.

In practice, CCC likely works even when this assumption doesn't strictly hold (e.g. in Pong and Coins), but with a slight modification to CCC we can use a much weaker condition: Let $\pi^{kC+\bar{k}D}$ be a policy that switches between $\pi^C$ and $\pi^D$ at most once every $k$ periods. Then we must only assume that there exists $k$ such that the payoff to agent 1 of $(\pi^{kC+kD}, \pi^C)$ is at least $\rho_{CC}$. In other words, as long as the Agent plays long enough stretches of $\pi^C$ or $\pi^D$ against $\pi^C$, their reward will be at least $\rho_C C$.

The modified CCC uses two thresholds: $T_D = (1 - \alpha_D)\rho_{CC} + \alpha_D\rho_{CD}$ is the threshold to switch from $\pi^C$ to $\pi^D$, and $T_C = (1 - \alpha_C)\rho_{CC} + \alpha_C\rho_{CD}$ is the threshold to switch back, with $\alpha_D > \alpha_C$. These thresholds grow linearly as $t \to \infty$ so the intervals between CCC switching policies has to grow linearly as well given bounded rewards.

## 6.3 FISHERY AND COINS TRAINING

The Fishery game has a $5 \times 10$ state space, where each agent is confined to, and observes, only their $5 \times 5$ half of the game. Coins is identical to Lerer & Peysakhovich (2017) but with the board size expanded to $8 \times 8$.

We follow the model architecture and training setup from Lerer & Peysakhovich (2017), with the modification that models are trained via policy gradient. We found that value-based methods (e.g. using a neural network baseline, or actor-critic) were unable to train prosocial policies in Fishery when using a stateless model. We suspect this is because states of very different prosocial value - 'I just ate the fish' vs. 'I let the fish swim to my partner' - are aliased in the observation space. Policy gradient with standard batch baselining produced converged policies for both Coins and Fishery. We trained Coins for 100,000 games and Fishery for 40,000 games.

For the tournament, agents play against opponents they have never trained against. The CCC statistics are computed at test time by rolling out a batch of 32 games for CC and 32 games for CD, updating the rollouts by one step per game step. Alternatively, the statistics (in particular, the defect threshold) could be pre-computed once for the longest game length of interest, and used in subsequent interactions.

## 6.4 PONG PLAYER DILEMMA TRAINING

We use the ALE environment modified for 2-player play as proposed in Tampuu et al. (2017), with modified rewards of +1 for scoring a point and -2 for being scored on.

We train policies directly from pixels, using the pytorch-a3c package `https://github.com/ikostrikov/pytorch-a3c`.

Policies are trained directly from pixels via A3C (Mnih et al., 2016). Inputs are rescaled to $42 \times 42$ and normalized, and we augment the state with the difference between successive frames. We use 38 threads for A3C, over a total of 38,000 games (1,000 per thread). We use the default settings from pytorch-a3c: a discount rate of $0.99$, learning rate of $0.0001$, 20-step returns, and entropy regularization weight of $0.01$.

The policy is implemented as a convolutional neural network with four layers, following pytorch-a3c. Each layer uses a $3 \times 3$ kernel with stride 2, followed by ELU. The network has two heads for the actor and critic. We elide the LSTM layer used in the pytorch-a3c library, as we found it to be unnecessary.

## 6.5 EXPERIMENT: COINS

We show that CCC maintains cooperation in the Coins game from Lerer & Peysakhovich (2017). In Coins two agents, Red and Blue, live on an $8 \times 8$ grid and move around in all cardinal directions[5].

---

[5]We use a larger grid size so the payoffs are not directly comparable to those reported in Lerer & Peysakhovich (2017)

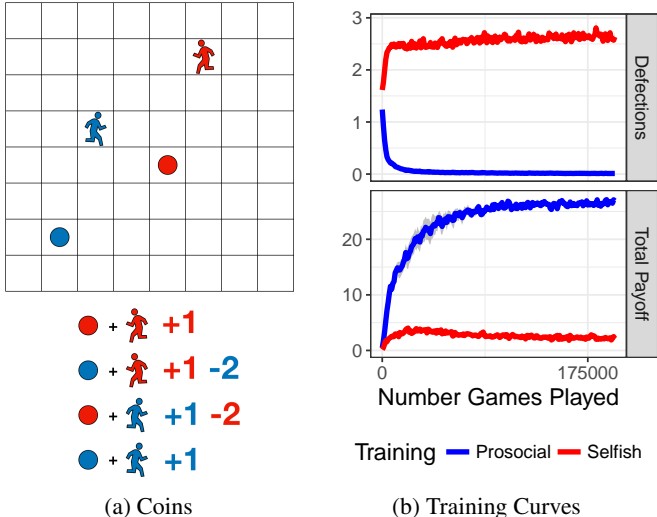

(a) Coins

(b) Training Curves

Figure 3: In Coins two agents, Red and Blue, live on an $8 \times 8$ grid and move around in all cardinal directions. Coins randomly appear on the board and each have a color. If an agent picks up (moves over) any coin, they receive one point. However, if they pick up a coin of the other agent's color, that agent loses 2 points. Selfish agents learn to grab all coins but prosocial agents play the strategy of only picking up coins of their own color. We see that CCC agents can avoid exploitation and maintain cooperation with cooperators and other CCC agents.

Coins randomly appear on the board and each have a color. If an agent picks up (moves over) any coin, they receive one point. However, if they pick up a coin of the other agent's color, that agent loses 2 points.

In Coins, the cooperative strategy is to pick up coins of one's color only. However, there is always a temptation to grab the other agents' coins. We train 10 copies of each strategy type under our two reward schemes and again compare payoffs of various policy combinations (see Figure 3). Note that Coins is a fully observed Markov game, however we still see that CCC (which has only limit guarantees and throws away all 'forward' information contained in an action) is just as effective as the intention based TFT strategies (which incentivize cooperation at all time periods) either built using $Q$ functions (Lerer & Peysakhovich, 2017) or learned via an end-to-end procedure (Foerster et al., 2017c).

## 6.6 TOURNAMENT RESULTS

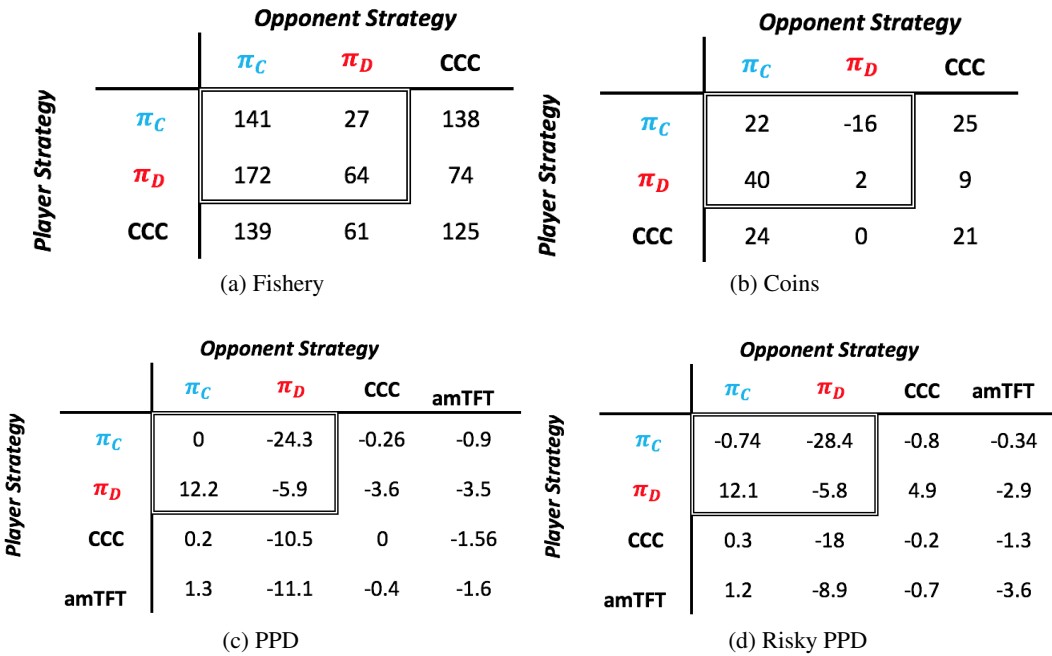

Figure 4: Tournament result for Fishery, Coins, PPD, and Risky PPD. Each cell shows the average total payoff of an agent playing the row strategy against a partner playing the column strategy.

