# OpenReview forum: "Consequentialist conditional cooperation in social dilemmas with imperfect information"
_ICLR.cc/2018/Conference — Accept (Poster)_

### Official Review · AnonReviewer2 · 2017-11-27
**Review on 'Consequentialist conditional cooperation in social dilemmas with imperfect information'**

**Rating:** 7
**Confidence:** 3

**Review:**

This paper proposes a novel adaptive learning mechanism to improve results in ergodic cooperation games. The algorithm, tagged 'Consequentialist Conditional Cooperation', uses outcome-based accumulative rewards of different strategies established during prior training. Its core benefit is its adaptiveness towards diverse opposing player strategies (e.g. selfish, prosocial, CCC) while maintaining maximum reward.

While the contribution is explored in all its technical complexity, fundamentally this algorithm exploits policies for selfish and prosocial strategies to determine expected rewards in a training phase. During operation it then switches its strategy depending on a dynamically-calculated threshold reward value (considering variation in agent-specific policies, initial game states and stochasticity of rewards) relative to the total reward of the played game instance. The work is contrasted to tit-for-tat approaches that require complete observability and operate based on expected future rewards. In addition to the observability, approximate Markov TFT (amTFT) methods are more processing-intense, since they fall back on a game's Q-function, as opposed to learned policies, making CCC a lightweight alternative.

Comments:

The findings suggest the effectiveness of that approach. In all experiments CCC-based agents fare better than agents operating based on a specific strategy. While performing worse than the amTFT approach and only working well for larger number of iterations, the outcome-based evaluation shows benefits. Specifically in the PPD game, the use of CCC produces interesting results; when paired with cooperate agents in the PPD game, CCC-based players produce higher overall reward than pairing cooperative players (see Figure 2, (d) & (e)). This should be explained. To improve the understanding of the CCC-based operation, it would further be worthwhile to provide an additional graph that shows the action choices of CCC agents over time to clarify behavioural characteristics and convergence performance.

However, when paired with non-cooperative players in the risky PPD game, CCC players lead to an improvement of pay-offs by around 50 percent (see Figure 2, (e)), compared to payoff received between non-cooperative players (-28.4 vs. -18, relative to -5 for defection). This leads to the question: How much CCC perform compared to random policy selection? Given its reduction of processing-intensive and need for larger number of iterations, how much worse is the random choice (no processing, independent of iterations)? This is would be worthwhile to appreciate the benefit of the proposed approach.

Another point relates to the fishing game. The game is parameterized with the rewards of +1 and +3. What is the bases for these parameter choices? What would happen if the higher reward was +2, or more interestingly, if the game was extended to allow agents to fish medium-sized fish (+2), in addition to small and large fish. Here it would be interesting to see how CCC fares (in all combinations with cooperators and defectors).

Overall, the paper is well-written and explores the technical details of the presented approach. The authors position the approach well within contemporary literature, both conceptually and using experimental evaluation, and are explicit about its strengths and limitations.

Presentation aspects:
- Minor typo: Page 2, last paragraph of Introduction: `... will act act identically.'
- Figure 2 should be shifted to the next page, since it is not self-explanatory and requires more context.

---

> ### Author Response · Authors · 2017-12-23
> **reply**
>
> We have made several additions to the paper that were suggested by the reviewer (the paper updated paper can be viewed via the PDF link above). We think these suggestions make the contribution more clear. We thank the reviewer for these comments.
>
> >>> Specifically in the PPD game, the use of CCC produces interesting results; when paired with cooperate agents in the PPD game, CCC-based players produce higher overall reward than pairing cooperative players (see Figure 2, (d) & (e)). This should be explained. **
>
> This is a good catch! Actually this is due to variance in the payoffs.
>
> The 2 cooperator payoff is -.74 with a standard error (calculated assuming that each matchup is independent so using the empirical standard deviation / sqrt(n-1)) of +/- .56 while the 2 CCC payoff is -.22 with a CI of  +/- .36. Thus, these payoffs are statistically indistinguishable.
>
> On the other hand 2 defectors get a payoff of -5.8 with a standard error of 2.2, so (D,D) is quite statistically distinguishable from (C,C) or (CCC,CCC).
>
> This stochasticity only occurs in the rPPD because of the random nature of the defect-payoff. A defection in the standard PPD means the partner loses 2 points deterministically whereas in the rPPD the partner only realizes a large loss of -20 with a relatively small probability of .1.
>
> In other words, the standard errors of the mean payoffs in the other games (eg. Fishery, standard PPD) are tiny and can be ignored but do need to be acknowledged in the rPPD.
>
> In the current version we have relegated the full tournament results to the appendix and edited what we show in the text to better reflect our problem definition.
>
> **>>> worthwhile to provide an additional graph that shows the action choices of CCC agents over time to clarify behavioural characteristics and convergence performance.**
>
> This is a good suggestion. We have added a figure that shows trajectories of behavior of a CCC agent with a C or D partner for the Fishery game.
>
> ** >>> How much CCC perform compared to random policy selection? **
>
> We note that while random policy selection will yield approximately a payoff of .5*(D,D) + .5*(C,D) when paired with a defect partner this random policy will no longer have the incentive properties of CCC are such that if our agent commits to CCC then their partner does better by cooperating than by defecting.
>
> By comparison, if our agent commits to a random policy this incentive property no longer holds and so the best response for a partner is to always defect.
>
> **The game is parameterized with the rewards of +1 and +3. What is the bases for these parameter choices? What would happen if the higher reward was +2, or more interestingly, if the game was extended to allow agents to fish medium-sized fish (+2), in addition to small and large fish. Here it would be interesting to see how CCC fares (in all combinations with cooperators and defectors). **
> ** **
> The choice of +1/ +3 was made rather arbitrarily (in most behavioral studies of cooperation the key question is one of the benefit/cost ratio, typically a ratio of 2:1 or 3:1 is used, we simply copied that here).
>
> We agree that more games are important for testing the robustness of CCC and other strategies for solving social dilemmas but we leave this for future work.
>
> **>> Typos/figure 2 clarity**
> ** **
> We thank the reviewer for these catches, we have fixed both of them.

---

> > ### Comment · AnonReviewer2 · 2018-01-12
> > **Thank you for the response**
> >
> > Thanks for clarifying some of the mentioned issues. With the introduced revision, you cover your ground very well. I believe that this paper offers a great basis for interesting further studies in this direction.

---

### Official Review · AnonReviewer1 · 2017-11-27
**Learning to cooperate with incomplete information**

**Rating:** 5
**Confidence:** 4

**Review:**

This paper studies learning to play two-player general-sum games with state (Markov games) with imperfect information. The idea is to learn to cooperate (think prisoner's dilemma) but in more complex domains. Generally, in repeated prisoner's dilemma, one can punish one's opponent for noncooperation. In this paper, they design an apporach to learn to cooperate in a more complex game, like a hybrid pong meets prisoner's dilemma game. This is fun but I did not find it particularly surprising from a game-theoretic or from a deep learning point of view.

From a game-theoretic point of view, the paper begins with a game-theoretic analysis of a cooperative strategy for these markov games with imperfect information. It is basically a straightforward generalization of the idea of punishing, which is common in "folk theorems" from game theory, to give a particular equilibrium for cooperating in Markov games. Many Markov games do not have a cooperative equilibrium, so this paper restricts attention to those that do. Even in games where there is a cooperative solution that maximizes the total welfare, it is not clear why players would choose to do so. When the game is symmetric, this might be "the natural" solution but in general it is far from clear why all players would want to maximize the total payoff.

The paper follows with some fun experiments implementing these new game theory notions. Unfortunately, since the game theory was not particularly well-motivated, I did not find the overall story compelling. It is perhaps interesting that one can make deep learning learn to cooperate with imperfect information, but one could have illustrated the game theory equally well with other techniques.

In contrast, the paper "Coco-Q: Learning in Stochastic Games with Side Payments" by Sodomka et. al. is an example where they took a well-motivated game theoretic cooperative solution concept and explored how to implement that with reinforcement learning. I would think that generalizing such solution concepts to stochastic games and/or deep learning might be more interesting.

It should also be noted that I was asked to review another ICLR submission entitled "MAINTAINING COOPERATION IN COMPLEX SOCIAL DILEMMAS USING DEEP REINFORCEMENT LEARNING" which amazingly introduced the same "Pong Player’s Dilemma" game as in this paper.

Notice the following suspiciously similar paragraphs from the two papers:

From "MAINTAINING COOPERATION IN COMPLEX SOCIAL DILEMMAS USING DEEP REINFORCEMENT LEARNING":
We also look at an environment where strategies must be learned from raw pixels. We use the method
of Tampuu et al. (2017) to alter the reward structure of Atari Pong so that whenever an agent scores a
point they receive a reward of 1 and the other player receives −2. We refer to this game as the Pong
Player’s Dilemma (PPD). In the PPD the only (jointly) winning move is not to play. However, a fully
cooperative agent can be exploited by a defector.

From "CONSEQUENTIALIST CONDITIONAL COOPERATION IN SOCIAL DILEMMAS WITH IMPERFECT INFORMATION":
To demonstrate this we follow the method of Tampuu et al. (2017) to construct a version of Atari Pong
which makes the game into a social dilemma. In what we call the Pong Player’s Dilemma (PPD) when an agent
scores they gain a reward of 1 but the partner receives a reward of −2. Thus, in the PPD the only (jointly) winning
move is not to play, but selfish agents are again tempted to defect and try to score points even though
this decreases total social reward. We see that CCC is a successful, robust, and simple strategy in this
game.

---

> ### Author Response · Authors · 2017-12-23
> **Reply**
>
> We thank the reviewer for their thorough review. We have made several changes in exposition and presentation. We hope these address the reviewer's concerns.
>
> >> The paper follows with some fun experiments implementing these new game theory notions. Unfortunately, since the game theory was not particularly well-motivated, I did not find the overall story compelling. It is perhaps interesting that one can make deep learning learn to cooperate with imperfect information, but one could have illustrated the game theory equally well with other techniques.
>
> From reading the reviewers' comments, we realize that we should have been much clearer front with our problem definition. We have edited the text substantially to make this clearer.
>
> Our goal is to consider a question of agent design: how should we build agents that can enter into social dilemmas and achieve high payoffs with partners that are themselves trying to achieve high payoffs?
>
> In particular, we seek to answer this question for social dilemmas where actions of a partner are not observed.
>
> This question is quite different from just making agents that cooperate all the time (since those agents are easily exploited by defectors). It is related to, but also different from, the question of computing cooperative equilibria.
>
> See the reply to R3 above for a more in depth discussion about the desiderata from past literature that we seek to satisfy in order to construct a “good” strategy for imperfect information social dilemmas.
>
> >> The reviewer asks whether maximizing the sum of the payoffs is the “right” solution
> We agree with this criticism. While in symmetric games (eg. bargaining) behavioral experiments show that often view the symmetric sum of payoffs to be a natural focal point while in asymmetric games they do not (see eg. the chapter on bargaining in Kagel & Roth Handbook of Experimental Economics or more recent work on inequality in social dilemmas eg. Hauser, Kraft-Todd, Rand, Nowak & Norton 2016).
>
> The question of how to choose the “correct” focal points for playing with humans comes down to asking what should the “right” utility function be for training the cooperative strategies (see Charness & Rabin (2002) for a generic utility function that can express many social goals). Note that CCC can then be applied using these new C strategies just as in the current work.
>
> However, figuring out the correct utility function to use here is far beyond the scope of this paper and is likely quite context dependent. This is an important direction for future research and we have made this point clear the paper.
>
>
> >> Similar paragraphs
> We are also the authors of the other paper, it is earlier/related work to this paper (in the sense that it asks about designing agents that can solve social dilemmas), though it covers substantially different ground (perfectly observed games).
>
> We apologize if there is something unclear from the current text. We do not mean to imply that this paper (CCC) introduces the PPD. Rather, it is the earlier paper (amTFT) that introduces the PPD as an environment and the CCC paper uses it for robustness checks.
>
> The point of the PPD in this paper is to ask whether the other work is superceded by the CCC. Indeed, the techniques proposed in the amTFT paper can ONLY work in perfectly observed games.
>
> By contrast, CCC is a good strategy for imperfectly observed games. Since any perfectly observed game is trivially also an imperfectly observed one one may think that CCC is just a strictly better strategy than amTFT (and thus the other paper is subsumed by this one).
>
> The point of the PPD experiments in this paper is to show that there are classes of perfectly observed games where CCC performs similarly to amTFT (normal PPD) but there are also those where CCC fails but amTFT succeeds (risky PPD).
>
> We have changed the text to make these points clearer and to attribute credit transparently.

---

### Official Review · AnonReviewer3 · 2017-11-28
**The paper makes a contribution to a challenging problem within multi-agent reinforcement learning. The paper is written clearly and it is easy to follow both the theoretical details and the general line of arguments. The paper would however, in my opinion, benefit from developing a number of areas of both the theoretical analysis and experimental section to solidify the contribution and validity.**

**Rating:** 6
**Confidence:** 4

**Review:**

The main result specifies a (trigger) strategy (CCC) and corresponding algorithm that leads to an efficient outcome in social dilemmas, the theoretical basis of which is provided by theorem 1. This underscores an algorithm that uses a prosocial adjustment of the agents rewards to encourage efficient behaviour. The paper makes a useful contribution in demonstrating that convergence to efficient outcomes in social dilemmas without the need for agents to observe each other's actions. The paper is also clearly written and the theoretical result is accompanied by some supporting experiments. The numerical experiments show that using CCC strategy leads to an increase in the proportion of efficient equilibrium outcomes. However, in order to solidify the experimental validation, the authors could consider a broader range of experimental evaluations. There are also a number of items that could be added that I believe would strengthen the contribution and novelty, in particular:

Some highly relevant references on (prosocial) reward shaping in social dilemmas are missing, such as Babes, Munoz de cote and Littman, 2008 and for the (iterated) prisoner's dilemma; Vassiliades and Christodoulou, 2010 which all provide important background material on the subject. In addition, it would be useful to see how the method put forward in the paper compares with other (reward-shaping) techniques within MARL (especially in the perfect information case in the pong players' dilemma (PPD) experiment) such as those already mentioned. The authors could, therefore, provide more detail in relating the contribution to these papers and other relevant past work and existing algorithms.

The paper also omits any formal discussion on the equilibrium concepts being used in the Markov game setting (e.g. Markov Perfect Equilibrium or Markov-Nash equilibrium) which leaves a notable gap in the theoretical analysis.

There are also some questions that to me, remain unaddressed namely:

i. the model of the experiments, particularly a description of the structure of the pong players' dilemma in terms of the elements of the partially observed Markov game described in definition 1. In particular, what are the state space and transitions?

ii. the equilibrium concepts being considered i.e. does the paper consider Markov perfect equilibria. Some analysis on the conditions that under which the continuation equilibria e.g. cooperation in the social dilemma is expected to arise would also be beneficial.

iii. Although the formal discussion is concerned with Markov games (i.e. repeated games with stochastic transitions with multiple states) the experiments (particularly the PPD) appear to apply to repeated games (this could very much be cleared up with a formal description of the games in the experimental sections and the equilibrium concept being used).

iv. In part 1 of the proof of the main theorem, it seems unclear why the sign of the main inequality has changed after application of Cauchy convergence in probability (equation at the top of the page). As this is an important component of the proof of the main result, the paper would benefit from an explanation of this step?

---

> ### Author Response · Authors · 2017-12-23
> **Reply**
>
> We thank the reviewer for their comments. They have pointed out weaknesses in our presentation of our results. We have edited the text substantially and hope that our contributions and claims are much clearer.
>
> **>>> R3 asks about the relationship of our work to prior work on reward shaping in MARL. **
>
> We are happy to add these references to the main text and discuss them. One important thing to note is that the prior work mentioned by the reviewer has dealt with perfectly observed games rather than partially observed ones.
>
> We have added a longer discussion of how our work is related to existing work on MARL, equilibrium finding, and reward shaping.
>
> We specifically discuss one of the examples the reviewer gives: Babes et al. 2008 use reward shaping in the repeated Prisoner's Dilemma to construct an agent that does well against fixed opponents as well as can lead learner “followers” to cooperate with it. However, in order to do this they first need to compute a value function for a known “good” strategy (they use Pavlov, a variant of tit-for-tat) and use this for shaping. This is possible for the basic one (or multi-memory) PD but doesn't scale well to general Markov games (in particular partially observed ones).
>
> By contrast, the CCC agent creates similar incentives by switching between two pre-computed strategies in a predictable way. The computation of these two strategies does not require anything other than standard self-play.
>
> Combining these ideas is an interesting direction for future research but beyond the scope of our paper.
>
> ** **
> **>> R3 asks for formalization for the state spaces/transition functions/etc… in our games.**
>
> We are happy to add more details of the games to the paper (as well as release the code upon publication). Our games do not permit a compact enumeration of the states and transitions (which is precisely why we are interested in moving beyond tabular methods to eg. deep RL). For example, in the PPD the full set of states is the set of RAM states in Atari Pong.
>
> **>> R3 asks about equilibrium concepts in our Markov game setting **
>
> While much existing work is framed in terms of finding good equilibria, our work is more related to questions raised by Axelrod (1984) who asks: if one is to enter a social dilemma with an (unknown) partner, how should one behave?
>
> The work on tit-for-tat (TFT, and related strategies such as Win-Stay-Lose-Shift/Pavlov) comes up with the answer that one should play a strategy that is:
>
>     * simple
>     * nice (begins by cooperating)
>     * not exploitable
>     * forgiving (provides a way to return to cooperation after a defection)
>     * incentivizes cooperation from its partner (that is, a partner who commits to cooperation will get a higher payoff than a partner who commits to defection)
>
> Our main contribution is to find a way to construct a strategy which satisfies the Axelrod desiderata in *partially observed* Markov games which require deep RL for function approximation.
>
> Note that these desiderata are different from equilibrium desiderata. For example, tit-for-tat, one of the most heavily studied strategies in the Prisoner's Dilemma, is actually not a symmetric equilibrium strategy (because the best response to TFT is always cooperate). Rather, these desiderata are about agent design or about good strategies to commit to.
>
> We do not claim that CCC forms an equilibrium with itself as there may be local deviations to improve one's payoff, however, our theorem shows that the partner of a CCC agent maximizes its asymptotic payoff by playing a policy that cooperates with the CCC agent (thus we preserve TFT-like incentive properties).
>
> We have edited the text to make our problem statement / results clearer.
>
> We only focus on equilibrium for computational reasons in the case of the D policy for which we have made an assumption that (D,D) forms an equilibrium in policies which only condition on agent's observations (this is related to the notion of a belief free equilibrium in repeated games Ely et al. 2005).
>
>
> **>> R3 asks “it seems unclear why the sign of the main inequality has changed after application of Cauchy convergence in probability (equation at the top of the page)”**
> ** **
> We apologize for any confusion. The equation at the top of the page uses the convention
>
> P(X) < epsilon
>
> while the next equations use the notation
>
> P(~X) > (1-epsilon)
>
> We have changed these to both use P(~X)>(1-epsilon) so that it is more clear.

---

### Decision · Program_Chairs · 2018-01-29
**ICLR 2018 Conference Acceptance Decision**

**Decision:**

Accept (Poster)

**Comment:**

The reviewer reactions to the initial manuscript were generally positive.  They considered the paper to be well written and clear, providing an original contribution to learning to cooperate in multi-agent deep RL in imperfect domains.  The reviewers raised a number of specific issues to address, including improved definitions and descriptions, and proper citations of related work.  The authors have substantially  revised the manuscript to address most or all of these issues.  At this point, the only knock on this paper is that the findings seemed unsurprising from a game-theoretic or deep learning point of view.

Pros: algorithmic contribution, technical quality, clarity
Cons: no real surprises